# Response to Abemaciclib and Immunotherapy Rechallenge with Nivolumab and Ipilimumab in a Heavily Pretreated TMB-H Metastatic Squamous Cell Lung Cancer with CDKN2A Mutation, PIK3CA Amplification and TPS 80%: A Case Report

**DOI:** 10.3390/ijms24044209

**Published:** 2023-02-20

**Authors:** Douglas Dias e Silva, Guilherme Bes Borba, Juliana Rodrigues Beal, Gehan Botrus, Akemi Osawa, Sérgio Eduardo Alonso Araújo, Fernando Moura, Rafael Aliosha Kaliks Guendelmann, Pedro Luiz Serrano Uson Junior

**Affiliations:** 1Department of Medical Oncology, Hospital Israelita Albert Einstein, Sao Paulo 05652-900, Brazil; 2Center for Personalized Medicine, Hospital Israelita Albert Einstein, Sao Paulo 05652-900, Brazil; 3Department of Oncology and Hematology, Emory University, Atlanta, GA 30322, USA; 4Department of Nuclear Medicine, Hospital Israelita Albert Einstein, Sao Paulo 05652-900, Brazil

**Keywords:** squamous cell lung cancer, lung cancer, nivolumab, ipilimumab, checkpoint inhibition therapy, abemaciclib, TMB-high

## Abstract

Inactivation of the cyclin-dependent kinase inhibitor 2A (CDKN2A) gene is considerably more frequent in squamous cell lung cancer (SqCLC) than in other subtypes of lung cancer and may be a promising target for this histology. Here, we present the course of diagnosis and treatment of a patient with advanced SqCLC, harboring not only CDKN2A mutation but also PIK3CA amplification, Tumor Mutational Burden-High (>10 mutations/megabase), and a Tumor Proportion Score of 80%. After disease progression on multiple lines of chemotherapy and immunotherapy, he responded favorably to treatment with the CDK4/6i Abemaciclib and later achieved a durable partial response to immunotherapy rechallenge with a combination of anti-PD-1 and anti-CTLA-4, nivolumab, and ipilimumab.

## 1. Introduction

Lung cancer survival has improved significantly, due to treatment advances in the last decade (e.g., targeted therapy and immunotherapy) [1]. Unfortunately, targeted therapy benefits most adenocarcinoma patients, and no molecular targets have been found specifically for squamous cell lung cancer (SqCLC) [2].

The cyclin-dependent kinase inhibitor 2A (CDKN2A) gene inactivation is significantly more common in SqCLC than in other non–small cell lung cancer (NSCLC) subtypes [3]. Thus, a better understanding of CDKN2A-associated alterations as oncogenic drivers may serve as a promising strategy for therapy with cyclin 4 and 6 inhibitors (CDK4/6i) [4,5,6,7]. The activity of CDK4/6i in the treatment of NSLC with or without CDKN2A mutations is not widely known, and there is limited single-agent efficacy [8,9,10,11,12,13,14].

The introduction of immune-checkpoint inhibitors (ICI) in the treatment of advanced NSCLC has transformed the therapeutic landscape of this disease [15]. Patients who have progressive disease in ICI usually receive chemotherapy upon progression. Nevertheless, a rechallenge may be attempted if progression occurs several months or years after the last dose of PD-L1 or programmed cell death protein 1 (PD-1) blockade [16,17].

Here, we present a case of a patient with advanced SqCLC that harbors not only CDKN2A mutation, and PIK3CA amplification, but also Tumor mutational burden High (TMB-H) and a tumor proportional score (TPS) of 50%. After disease progression on multiple lines of chemotherapy and ICI, treatment with Abemaciclib was attempted. Although a stable disease with metabolic response was achieved, the patient also developed grade 4 interstitial lung disease (ILD) and had to discontinue therapy. After the resolution of this adverse event, he achieved a durable response to ICI rechallenge with a combination of anti-PD-1 and anti-CTLA-4.

## 2. Case Description

A 60-year-old male former smoker presented with dyspnea during physical activity in November 2015. Computed tomography (CT) scans of the chest revealed a mass in the left upper lobe, measuring 5.8 cm; paralysis of the left diaphragm; and an enlarged para-aortic lymph node. Transbronchial biopsy revealed SqCLC. He underwent left upper lobectomy with lymphadenectomy, confirming an SqCLC pT3pN0 (stage IIB per the 8th edition AJCC/TNM), although it was deemed an R1 resection due to the involvement of the mediastinal fat. Immunohistochemistry showed high expression of PD-L1, with a TPS of 50%, assessed by DAKO 22C3^®^. A detailed treatment overview is depicted in Figure 1.

He received adjuvant chemotherapy with Cisplatin 50 mg/m^2^ intravenous (IV) D1 and D8 Vinorelbine 25 mg/m^2^ IV D1, D8, D15, D22 q4w for four cycles, followed by image-guided radiation therapy (IGRT) of 66 Gy due to mediastinal fat involvement. After 6 months of completion of adjuvant treatment, he developed mediastinal recurrence. He received Pembrolizumab 200 mg IV q3w as first-line therapy starting in September 2017 and upon progression, in a single parasternal lesion, he was maintained on Pembrolizumab and underwent hypofractionated stereotactic radiotherapy (SBRT) with a total dose of 25 Gy in January 2018. In November 2018, when the new progression of disease in soft tissue and lymph nodes in the internal thoracic chain with extension to the anterior mediastinal fat and pericardial thickening was documented, he underwent surgical resection.

Second-line chemotherapy with Carboplatin AUC 6 plus Paclitaxel 200 mg/m^2^ for four cycles was started on January 2019, due to a new mediastinal progression, achieving a partial response. From June 2019 to November 2020, the patient received multiple lines of chemotherapy (Cisplatin 75 mg/m^2^ IV D1 and Gemcitabine 1000 mg/m^2^ D1, D8 q3w, Docetaxel 75 mg/m^2^ plus Ramucirumab 10 mg/kg IV q3w, Cisplatin 100 mg/m^2^ D1 plus Vinorelbine 25 mg/m^2^ D1, D8, D15, D22 and Nab-Paclitaxel 100 mg/m^2^ IV q3w) due to subsequent local disease progressions (lymph nodes and subcutaneous). Clinical benefit and initial response were observed; however, after a few months on each line, the disease eventually progressed. In September 2020, a somatic panel (TARGET^®^) was performed, revealing TMB of 10.19 mutations/Megabase and the following clinically relevant genetic variants: Gene *CDKN2A*, variant: c.116delA p.(Asn39Thrfs*14), gene: *PIK3CA*, variation: amplification, gene *ARID1A*, variant: c.971dupG p.(Ala325Argfs*75) and *TP53* variant: c.820G>T p.(Val274Phe).

In November 2020, new images documented a new anterior chest wall tumor and disease progression to the sternum and pericardium. Considering multiple prior lines of treatment including chemotherapy and immunotherapy, a good performance status (ECOG 1), case reports and a phase II trial of the benefit of CDK4/6i Palbociclib in tumors which harbor somatic alterations in *CDKN2A* and, after informed consent, off-label Abemaciclib was started on December 2020. Re-staging PET-CT on February 2021, after 8 weeks of treatment with good tolerance, showed stable disease with metabolic response in the pericardial lesion, costochondral joints, and sternum (Figure 2).

Treatment was maintained until March 2021, when the patient developed dyspnea and respiratory failure secondary to acute pneumonitis requiring orotracheal intubation. High-resolution CT and bronchoalveolar lavage were performed. Grade 4 pneumonitis related to Abemaciclib was considered the most likely diagnosis after the exclusion of infection and disease-related complications. He eventually improved with corticosteroid and Abemaciclib suspension (Figure 3).

In July 2021, after full recovery from pneumonitis and having remained with no cancer treatment for 3 months and still having measurable disease, an immunotherapy rechallenge was recommended. He received two cycles of Ipilimumab 1 mg/kg plus Nivolumab 3 mg/kg, followed by maintenance with Nivolumab 240 mg q2w. In January 2022, a new PET-CT showed a partial response and complete metabolic response to immunotherapy rechallenge (Figure 4). The patient is still on maintenance nivolumab, sustaining a durable response confirmed by a PET-CT performed in October 2022.

## 3. Discussion

Within the last decade, the use of genomic profiling shed light on targetable mutations, allowing for personalized therapies for specific oncogenic drivers [1]. Unfortunately, no such molecular targets have been identified for SqCLC, resulting in poorer treatment outcomes in patients with advanced disease [2]. Additional research is necessary to identify and select the appropriate treatment for patients with SqCLC according to their molecular features [4,13].

The CDKN2A gene is frequently inactivated in human cancers, including NSCLC. The Cancer Genome Atlas (TCGA) project revealed that CDKN2A is inactivated in 72% of SqCLC [3]. CDKN2A is a gene located on chromosome 9, which contains two introns, and three exons and encodes the p16INK4a and p14ARF proteins that inhibit the formation of the complex between CDK4/6 and cyclin D, protects p53 from being broken down, and induces the expression of p21, a CDK inhibitor [4,18,19]. The CDKN2A gene also activates the retinoblastoma (Rb) family of proteins, which block the cell from progressing from the G1 to the S phase [20]. The expression of CDK2/4/6 is increased in CDKN2A-defective cells; therefore, CDK4/6i could be a valuable mechanism to prevent cell cycle progression and thus induce cell senescence [5]. The activity of CDK4/6i in metastatic NSCLC is yet to be fully comprehended and explored. Phases I, II and III trials were conducted using CDK4/6i in this setting, with suboptimal drug activity (Table 1).

Our patient also had a PIK3CA amplification and TP53 mutation. *CDKN2A* and *PI3K* signaling pathways modulate the activity of Cyclin D, an important protein involved in regulating cell cycle progression and DNA synthesis. The presence of both *CDKN2A*, *PI3K* and TP53 mutations may justify a greater efficiency of the CDK4/6i Abemaciclib in patients who present these combined alterations [7,19,21,22,23], as reported in this case (Figure 5).

Albeit leading to significant clinical, radiologic, and metabolic response, Abemaciclib, after 8 weeks of treatment, had to be withheld due to grade 4 pneumonitis. This is a rare adverse event. Data from MONARCH trials 2 and 3 showed that drug-induced lung injury of any grade occurred in 3.4%, with grade 3 events occurring in ≤1% of cases [24].

Data has emerged more recently linking alterations in *CDKN2A*, with ICI resistance in several solid tumors, including NSCLC [25]. However, our patient had a good initial response with Pembrolizumab and later with Ipilimumab plus Nivolumab rechallenge. The significant response to ICIs despite the CDKN2A mutation could be explained by the high expression of PD-L1 and TMB-High, in addition to prior exposure to Abemaciclib.

Chemoimmunotherapy and immunotherapy alone have led to improved outcomes in NSCLC, although limited data about treatment-related biomarker-driven responses are available [26,27,28]. The KEYNOTE-024 trial showed that pembrolizumab was associated with longer PFS and OS compared to platinum-based combination chemotherapy in patients with previously untreated advanced NSCLC and a PD-L1 tumor proportion score of 50% or greater [29].

The association of TMB-H with outcomes was assessed in phase 2 of the KEYNOTE-158 study, which evaluated the anti-PD-1 monoclonal antibody pembrolizumab in patients with previously treated, advanced solid tumors. The study used a prespecified cut point of at least 10 mutations per megabase to define TMB-High tumors, identifying a subgroup of patients who could have a robust tumor response to pembrolizumab monotherapy [30]. A phase III trial showed PFS significantly longer with the first-line combination of PD-1 and CTLA-4 nivolumab plus ipilimumab versus chemotherapy among patients with stage IV or recurrent NSCLC that was not previously treated with chemotherapy and a TMB-H (≥10 mutations per megabase) [31].

Most patients who initially benefited from ICI will eventually become resistant [32]. Acquired resistance to immunotherapy is defined by a measured objective response to therapy, followed by progression on or after the therapy [33,34]. In melanoma, after a disease progression to checkpoint inhibitors, patients usually receive other therapies, adding an ICI of a different class. The interval without immunotherapy might sensitize cancer to a subsequent checkpoint inhibition exposition [34,35]. If progression occurs several months or years after the last dose, rechallenge may still be an option, even in lung cancer [17]. The lack of tumor antigens, defective antigen presentation, modulation of critical cellular pathways, epigenetic changes, and changes in the tumor microenvironment are all examples of resistance mechanisms that have been well-documented [36].

To our knowledge, this is the first report of immunotherapy rechallenge in a heavily pre-treated patient with squamous cell lung cancer after drug-induced pneumonitis. Furthermore, the durable tumor response observed in this case associated with the low incidence of new adverse events could represent a new strategy to be further explored in larger cohorts.

## 4. Conclusions

Therapy with Abemaciclib demonstrated metabolic, radiologic, and clinical response in a heavily pretreated patient with metastatic SqCLC harboring CDKN2A and TP53 mutation in addition to *PIK3CA* amplification. Immunotherapies rechallenge with anti-PD1/anti-CTLA4 doublet achieved a durable radiologic response in the setting of TPS 80%. Reports, even of single cases, may lead to new treatment strategies based on common driver mutations. Rechallenges with immunotherapy should be prospectively studied to identify predictors of clinical benefit.

## Figures and Tables

**Figure 1 ijms-24-04209-f001:**
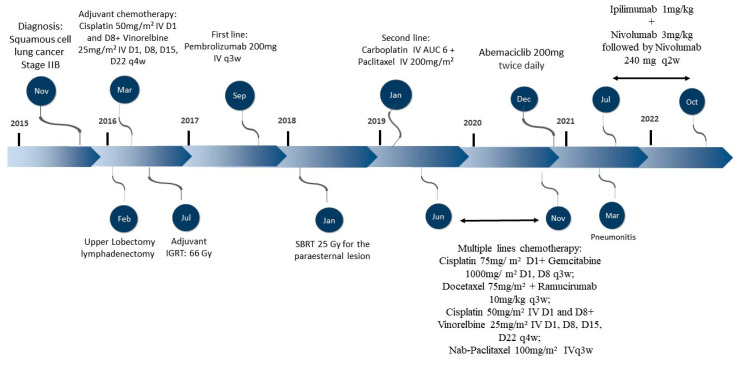
Overview of the patient’s course of disease and treatment regimens.

**Figure 2 ijms-24-04209-f002:**
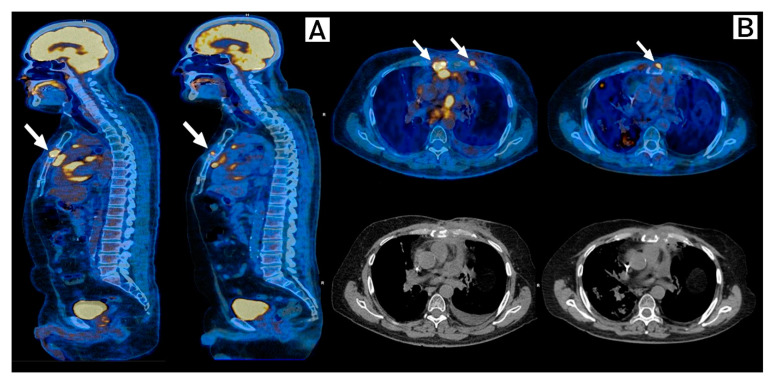
Response to Abemaciclib in the sternum and costochondral lesions. FDG-PET-Scan, Baseline (**left**) and 8 weeks after (**right**) after initiation of Abemaciclib. (**A**) Sagittal plane and (**B**) Transversal plane. Arrows show stable disease and complete metabolic resolution of costochondral lesions and near-complete metabolic response of the sternal lesion when compared with prior scans.

**Figure 3 ijms-24-04209-f003:**
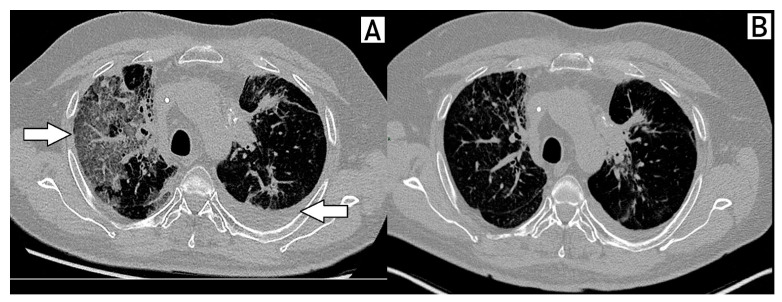
Pneumonitis associated with abemaciclib. Arrows in the figure shows ground-glass opacities, suggesting pneumonitis of the right lung and pleural effusion in the left lung (**A**). Resolution of pneumonitis 3 months after drug discontinuation (**B**).

**Figure 4 ijms-24-04209-f004:**
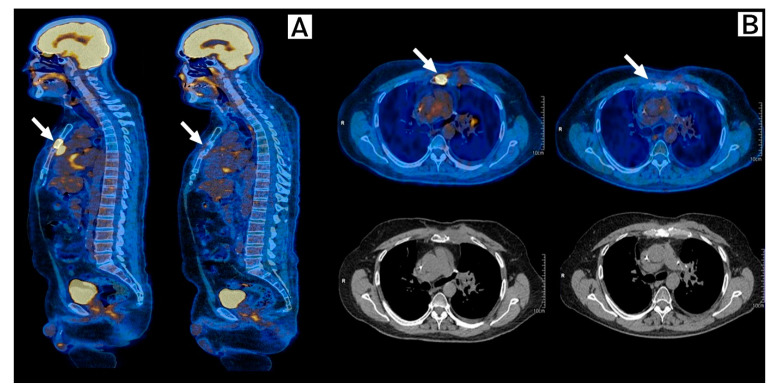
Response to immunotherapy rechallenge. FDG-PET-Scan, Baseline (**left**) and 6 months after (**right**) after initiation of Ipilimumab plus Nivolumab. (**A**) Sagittal plane, and (**B**) Transversal plane. Arrows show a partial response with metabolic response of sternum and costochondral lesions when compared to prior scans, demonstrating a response to Ipilimumab plus Nivolumab.

**Figure 5 ijms-24-04209-f005:**
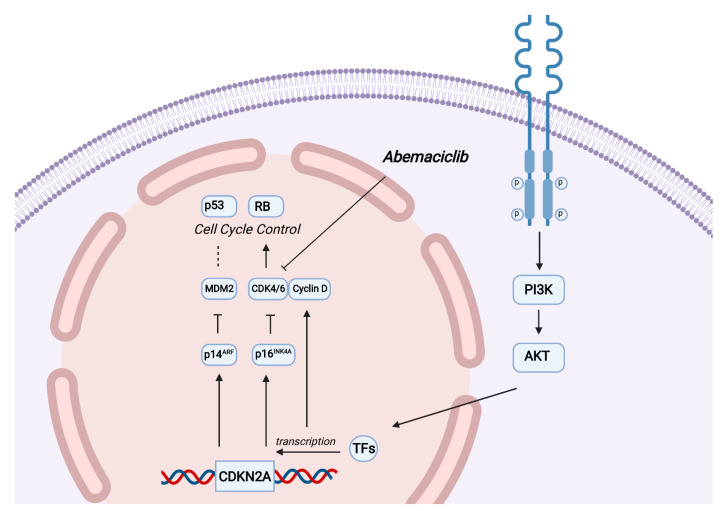
Abemaciclib effect on CDKN2A and PI3K mutated cancer cells. CDKN2A encodes the tumor suppressors p16ink4a, and p14ARF. PI3K-AKT pathway controls several transcription factors (TFs), leading to an increase in Cyclin D expression. Abemaciclib prevents cell cycle progression, inhibiting the complex CDK4/cyclin D, responsible for the phosphorylation and inhibition of the RB tumor-suppressor gene.

**Table 1 ijms-24-04209-t001:** Summary of clinical trials, including data on CDK4/6i in NSCLC.

Author, Year	Therapy	Phase	N	Molecular Analysis	Response to CDKi
Goldman et al., 2014 [8]	Abemaciclib	I	49	KRAS	DCR: 54% for KRAS mut
Gopalan et al., 2018 [9]	Palbociclib	II	16	p16 null	SD: 50% lasting 4–10 m
Goldman et al., 2018 [10]	Abemaciclib vs. Erlotinib	III	453	KRAS mut. (Codons 12 and 13)	mOS: 7.4 m vs. 7.8 m mPFS: 3.6 m vs. 1.8 m DCR: 54% vs. 31% ORR: 8.9% vs. 2.7%
Scagliotti et al., 2018 [11]	Abemaciclib vs. Docetaxel	II	159	NMA	mOS: 7 m vs. 12.4 m mPFS: 2.5 m vs. 4.2 m DCR: 50% vs. 64% ORR: 2.8% vs. 4.2%
Kim et al., 2018 [12]	Abemaciclib plus Pemetrexed, Gemcitabine or Ramucirumab	Ib	86	NMA	DCR: 52% for pemetrexed group DCR: 25% for gemcitabine group DCR: 54% for Ramucirumab group
Edelman et al., 2019 [13]	Palbociclib	II	32	CDK4, CCND1, CCND2, CCND3 amplifications	mOS: 7.1 m mPFS: 1.7 m ORR: 6% SD: 38%
Ahn et al., 2020 [14]	Palbociclib	II (TAPUR Studt)	29	CDKN2A alterations/No Rb mutations	mOS: 21.6 w mPFS: 8.1 w DCR: 31%

Abbreviations: DCR, disease control rate; SD, stable disease; mOS median overall survival; mPFS, median progression-free survival; ORR, overall response rate; VS, versus; NMA, no molecular analysis; TAPUR, The Targeted Agent and Profiling Utilization Registry; M, months; W, weeks; mut, mutant.

## Data Availability

Not applicable.

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
