# Peer review of "Response to Abemaciclib and Immunotherapy Rechallenge with Nivolumab and Ipilimumab in a Heavily Pretreated TMB-H Metastatic Squamous Cell Lung Cancer with CDKN2A Mutation, PIK3CA Amplification and TPS 80%: A Case Report"

_ijms, 2023, doi:10.3390/ijms24044209_

Round 1

Reviewer 1 Report

Manuscript has a good quality and I recommend it for publication after major correction.

-          More explanations are needed for improving of the paper novelty.

-          Abbreviations should probably be in the introduction.  Abbreviations which are not repeated must not be a part of abstract.

-          Please add the several sentences to describe significance for this system.

-          many places use repeated names, please just use the full name once and then the acronyms

-          Introduction: some parts are too long, and must be shortened, please do not repeat sentences or facts.

-          Abstract: it must be shortened; Please refer to what is proper in this new research.

Author Response

Author's Reply to the Review Report (Reviewer 1)

Thank you for the review

-          More explanations are needed for improving of the paper novelty

Answer: Thank you for the suggestion. We have now expanded the novelty status of the manuscript, in the discussion.

-          Abbreviations should probably be in the introduction.  Abbreviations which are not repeated must not be a part of abstract.

Answer: Thank you for the suggestion. We have now made changes to abbreviations to avoid unnecessary repetition in the text.

-          many places use repeated names, please just use the full name once and then the acronyms.

Answer: Thank you for the suggestion. We have now excluded repeated names and kept only acronyms as they appeared in the text

-          Introduction: some parts are too long, and must be shortened, please do not repeat sentences or facts.

Answer: Thank you for the suggestion. We have now shortened the introduction as suggested by trying to highlight the most importante information concisely and avoiding repetition

-          Abstract: it must be shortened; Please refer to what is proper in this new research.

Answer: Thank you for the suggestion. We have now shortened the abstract highlighting what is most appropriate for this case report

Reviewer 2 Report

Nice presented and clear work with relevant information to the medical community. 

Author Response

Author's Reply to the Review Report (Reviewer 2)

Thank you very much for the review

Comments and Suggestions for Authors

Nice presented and clear work with relevant information to the medical community. 

Answer: Thank you for the important insight about the manuscript. We addressed future applications of the treatments performed in this case report in the discussion.

Reviewer 3 Report

This case report titled “Response to abemaciclib and immunotherapy rechallenge with nivolumab and ipilimumab in a heavily pretreated TMB-H metastatic squamous cell lung cancer with CDKN2A mutation, PIK3CA amplification and TPS 80%: A Case report”.

The entire case report is well written. This paper can be published after minor revision.

Comments:

1, The description of the patient’s course of disease and treatment regimens in main text are inconsistent to Figure 1, e.g.:

“25 Gy in January 2018” should move from 2017 to 2018 in Figure 1;

“Second-line chemotherapy with Carboplatin AUC 6 plus Paclitaxel 200 mg/m2” should move from 2018 to 2019 in Figure 1;

“Cisplatin 100 mg/m2 plus 88 Vinorelbine 30 mg/m2” in manuscript, but it is “Cisplatin 50 mg/m2 plus 88 Vinorelbine 25 mg/m2” in Figure 1, the authors should clarify this treatment.

2, Does this patient have abnormal EGFR or ALK gene?

Author Response

Author's Reply to the Review Report (Reviewer 3)

Thank you very much for the review

Comments:

1, The description of the patient’s course of disease and treatment regimens in main text are inconsistent to Figure 1, e.g.:

“25 Gy in January 2018” should move from 2017 to 2018 in Figure 1;

Answer: Thank you for your suggestion. We have now corrected the Figure 1.

“Second-line chemotherapy with Carboplatin AUC 6 plus Paclitaxel 200 mg/m2” should move from 2018 to 2019 in Figure 1;

Answer: Thank you for your suggestion. We have now changed the Figure 1. We have moved “Second-line chemotherapy with Carboplatin AUC 6 plus Paclitaxel 200 mg/m2” from 2018 to 2019.

Cisplatin 100 mg/m2 plus 88 Vinorelbine 30 mg/m2” in manuscript, but it is “Cisplatin 50 mg/m2 plus 88 Vinorelbine 25 mg/m2” in Figure 1, the authors should clarify this treatment.

Answer: Thank you for your suggestion. We have now corrected Figure 1. The correct treatment protocol is Cisplatin 50mg/m2 D1 plus Vinorelbine 25mg/mD1,D8,D15,D22. I have added a new “Figure 1” with the changes that were suggested.

2, Does this patient have abnormal EGFR or ALK gene?

Answer: Thank you very much for your review. This patient was evaluated with a somatic panel that identified no actionable abnormalities in either ALK or EGFR.

Reviewer 4 Report

This case report by Dias e Silva et al. is a great example of the implications of genomic medicine in oncology. The case report is clear, sound and in good standing. However, I have a few suggestions:

1. Authors should highlight their approach in accordance to the guidelines - a figure could be useful.

2. Abbreviations need to be in order - eg TPS

3. line 27 - should read “respectively” to indicate which drug is which.

4. line 60 - anti-CTLA4

Best regards

Author Response

Author's Reply to the Review Report (Reviewer 4)

Thank you very much for the review.

1.Authors should highlight their approach in accordance to the guidelines - a figure could be useful.

Answer: Thank you for your suggestion. We have included our institutional guideline for moecular testing and treatment for advanced non-small cell lung cancer as supplementary material to this manuscript.

  1. Abbreviations need to be in order - eg TPS

Answer: Thank you for the suggestion. We have now reviewed and modified all the abbreviations to make them concordant. Thank you.

  1. line 27 - should read “respectively” to indicate which drug is which.

Answer: Thank you for the suggestion. We have now changed the sentence to the following “…rechallenge with a combo of anti-PD-1 and anti-CTLA-4, nivolumab and ipilimumab.

  1. line 60 - anti-CTLA4

Answer: Thank you for pointing this out. We have now corrected the sentence to the following: “… he achieved a durable response to ICI rechallenge with a combination of anti-PD-1 and anti-CTLA-4.